# Tumoricidal Activity of Simvastatin in Synergy with RhoA Inactivation in Antimigration of Clear Cell Renal Cell Carcinoma Cells

**DOI:** 10.3390/ijms24119738

**Published:** 2023-06-04

**Authors:** Yuan-Chii Gladys Lee, Fang-Ning Chou, Szu-Yu Tung, Hsiu-Chu Chou, Tsui-Ling Ko, Yang C. Fann, Shu-Hui Juan

**Affiliations:** 1Graduate Institute of Biomedical Informatics, College of Medical Science and Technology, Taipei Medical University, Taipei 11031, Taiwan; ycgl@tmu.edu.tw; 2Department of Physiology, School of Medicine, College of Medicine, Taipei Medical University, Taipei 11031, Taiwan; carcatqqq@gmail.com (F.-N.C.); sayeodong@gmail.com (S.-Y.T.); 3Department of Anatomy and Cell Biology, School of Medicine, College of Medicine, Taipei Medical University, Taipei 11031, Taiwan; chou0217@tmu.edu.tw; 4College of Science, National Sun Yat-Sen University, Kaohsiung 80424, Taiwan; kate819b@gmail.nsysu.com; 5Intramural IT and Bioinformatics Program, Division of Intramural, National Institute of Neurological Disorders and Stroke, National Institutes of Health, Bethesda, MD 20892, USA; fann@ninds.nih.gov

**Keywords:** renal cell carcinoma cells, Simvastatin, RhoA, mevalonate, lipid deposition, apoptosis

## Abstract

Among kidney cancers, clear cell renal cell carcinoma (ccRCC) has the highest incidence rate in adults. The survival rate of patients diagnosed as having metastatic ccRCC drastically declines even with intensive treatment. We examined the efficacy of simvastatin, a lipid-lowering drug with reduced mevalonate synthesis, in ccRCC treatment. Simvastatin was found to reduce cell viability and increase autophagy induction and apoptosis. In addition, it reduced cell metastasis and lipid accumulation, the target proteins of which can be reversed through mevalonate supplementation. Moreover, simvastatin suppressed cholesterol synthesis and protein prenylation that is essential for RhoA activation. Simvastatin might also reduce cancer metastasis by suppressing the RhoA pathway. A gene set enrichment analysis (GSEA) of the human ccRCC GSE53757 data set revealed that the RhoA and lipogenesis pathways are activated. In simvastatin-treated ccRCC cells, although RhoA was upregulated, it was mainly restrained in the cytosolic fraction and concomitantly reduced Rho-associated protein kinase activity. RhoA upregulation might be a negative feedback effect owing to the loss of RhoA activity caused by simvastatin, which can be restored by mevalonate. RhoA inactivation by simvastatin was correlated with decreased cell metastasis in the transwell assay, which was mimicked in dominantly negative RhoA-overexpressing cells. Thus, owing to the increased RhoA activation and cell metastasis in the human ccRCC dataset analysis, simvastatin-mediated Rho inactivation might serve as a therapeutic target for ccRCC patients. Altogether, simvastatin suppressed the cell viability and metastasis of ccRCC cells; thus, it is a potentially effective ccRCC adjunct therapy after clinical validation for ccRCC treatment.

## 1. Introduction

Renal cell carcinoma (RCC) is the most common type of kidney cancer (90%), and clear cell RCC (ccRCC) has the highest prevalence rate (70%) in adults. Studies have demonstrated that the 5-year survival rate for kidney cancer is approximately 75%, whereas the survival rate for metastatic cancer has sharply declined to only 12% [1]. Patients with RCC are generally resistant to chemotherapy, with only a small percentage of patients responsive to cytokine treatment [2]. Nephrectomy is the common and effective treatment for such patients, although the prognosis of RCC is mainly related to the clinical tumor stage. Targeted therapy includes sorafenib tosylate (Nexava) and temsirolimus (Torisel), which are oral inhibitors of multiple kinases (e.g., VEGFs and PDGF and mammalian target of rapamycin (mTOR), respectively). However, the efficacy of such targeted therapy is insufficient in patients with metastatic RCC. Thus, new therapeutic targets for antimetastasis must be urgently identified.

Simvastatin is clinically used to treat patients with hyperlipidemia and hypercholesterolemia. It works by lowering blood lipid levels and blocking the synthesis of cholesterol through the inhibition of hydroxyl-methyl-glutaryl-coenzyme A (HMG-CoA) reductase, a key enzyme in cholesterol synthesis. In ccRCC, simvastatin exerts apoptotic and antimetastatic effects on cancer cells by inhibiting the AKT/mTOR, ERK, and JAK2/STAT3 pathways [3]. Wang et al. [4] demonstrated that simvastatin potentially delays cancer cell progression by initiating apoptosis and inducing cell cycle arrest in the G0/G1 phase. However, in patients who receive a diagnosis in potential metastasis stages, the apoptosis and cell cycle arrest effects of a drug might not be effective. Numerous studies have reported that statins can attenuate breast cancer cell metastasis [5]. Nevertheless, research on the antimetastasis effect of simvastatin in ccRCC is limited.

ccRCC is a kidney tumor that originates from renal epithelium cells. It usually presents as a solid, fat-like cyst, and these cancer cells have a clear cytoplasm and relatively small nuclei. Studies have determined that the clear cytoplasm is mainly composed of lipids and glycogen [6]. Sterol responsive element binding factor (SREBF), a downstream transcription factor of the mTOR pathway, modulates the expression of metabolic related genes, including those for acetyl-CoA carboxylase (ACC), fatty acid synthase (FASN), and stearoyl-CoA desaturase (SCD). The accumulated lipids have a protective effect on ccRCC. Thus, investigating the contributions of metabolic abnormalities to ccRCC progression and whether the abnormalities can be reversed can provide insights to develop new therapeutic strategies.

The activation of Rho GTPases (e.g., RhoA) regulates cytoskeletal rearrangement, cell contraction, cell migration, and focal adhesion. RhoA is upregulated in various human cancers such as ovarian, gastric, and testicular cancers [7,8,9]. RhoA upregulation stimulates cell cycle progression and migration in mouse models [6,7]. Rho-associated protein kinase (ROCK), a downstream target of RhoA, modulates cytoskeleton and cell morphology and is critical for endothelial cell function in normal and pathologic states. Inhibition of ROCK has been demonstrated to significantly block VEGF-mediated endothelial cell differentiation and activation in cancer studies [8,9]. RCC is a highly vascularized neoplasm; thus, antiangiogenesis through ROCK inhibition might attenuate RCC progression. In this study, we characterized the molecular mechanisms underlying the anti-ccRCC effects of simvastatin on cell viability, lipid metabolism, and cell metastasis and identified the proteins involved in autophagy, apoptosis, lipid synthesis, and cytoskeletal regulation.

## 2. Results

### 2.1. Molecular Mechanisms Underlying the Antitumor Effect of Simvastatin on ccRCC Cells

We evaluated the potential anti-ccRCC effects of simvastatin. The results of the MTT analysis revealed that simvastatin inhibited cell viability in various ccRCC cells, including Caki1, Caki2, RCC52, 786O (VHL wild-type), and A498 (VHL mutant) cells in a concentration-dependent manner; moreover, cell numbers were significantly decreased after treatment with simvastatin (1–10 μM) for 24–72 h (Figure 1A). After 48 h treatment with simvastatin (2.5 μM), cell cycle arrest was induced in the G0/G1 phase, and the subG0/G1 peak was increased, as revealed by flow cytometry analysis (Figure 1B). Additionally, cell apoptosis was evidenced by the increase in the DNA ladder and the number of apoptotic bodies by DAPI staining (Figure 1C), and early and late apoptosis was visualized by double staining with annexin V-FITC/PI (Figure 1D).

The antimigratory effect of simvastatin was assessed in migrated cells stained by crystal violet dye in the transwell assay (Figure 2A), and the results revealed that 1–5 μM simvastatin reduced Caki2 cell invasion in a concentration-dependent manner. To examine whether simvastatin reduced lipid accumulation, lipid droplets in ccRCC cells were stained with ORO. The results revealed that simvastatin reduced lipid accumulation in Caki2 cells in a concentration-dependent manner (Figure 2B). Furthermore, the potential molecular mechanism underlying the antitumor effect of simvastatin was assessed in A498 cells through Western blot analysis (Figure 2C), which revealed that simvastatin (1–10 μM) concentration-dependently increased the expression levels of phospho-liver kinase B1 (pLKB1), phospho-AMP-activated protein kinase alpha (pAMPKα), RhoA, p21/p27, microtubule-associated protein 1 light chain 3B II (LC3BII, an autophagy marker), and cleaved caspase 3 and reduced the levels of FASN, SCD (lipogenesis), phospho-protein kinase B1 (pAKT1), pmTOR, and cyclin D1/D3. Thus, we hypothesize that simvastatin might alleviate ccRCC progression in cell proliferation. To assess the efficacy of simvastatin in vivo, simvastatin was given twice a week for 4 weeks to nude mice with xenograft transplantation. The results in Figure 2D show that tumor weights (gw) and volumes (cm^3^) of subcutaneous xenografts grown from Caki2 cells were significantly smaller than those from additional simvastatin treatment (*p* < 0.05). The values of BUN in control and simvastatin groups were 18 ± 2 and 20 ± 2 mg/dL, and those of GPT were 45 ± 3 and 47 ± 2 U/L, respectively, which were within the normal ranges and no statistical significance was determined between groups.

### 2.2. Anticancer Effects of Simvastatin through the Inhibition of the Mevalonate Pathway

Because simvastatin inhibited mevalonate synthesis by inhibiting HMG-CoA reductase, the contribution of mevalonate to the antitumor effects of simvastatin was examined through mevalonate supplementation. The results of the Western blot analysis in Figure 3A indicate that mevalonate supplementation could significantly reverse the altered levels of proteins involved in the bioenergetic metabolism (e.g., pLKB1, pAMPKα, pmTOR, FASN, and SCD) of pAKT involved in cell growth/survival signaling and of proteins involved in cell adhesion/cytoskeletal organization (e.g., RhoA) and cell cycle/autophagy/apoptosis (e.g., cyclin D1/D3, p21/p27, LC3BII, and cleaved PARP). In accordance with the reversal of the effects of simvastatin by mevalonate at the molecular levels, mevalonate supplementation also attenuated the reduced cell viability and increased the apoptosis caused by simvastatin in an apoptotic flow analysis with annexin v-FITC/PI double staining and the MTT assay (Figure 3B,C).

### 2.3. Activation of RhoA Signaling and the Fatty Acid Synthesis Pathway in the GSEA Analysis of the Human ccRCC Data Set

We selected the GSE53757 human data set composed of samples of 72 patients with ccRCC, with paired normal and ccRCC tumor tissues, of which one paired sample was an outlier and hence was removed after data normalization. The patients were grouped into different TNM stages (stages 1–4) based on their pathological features. Gene expression profiles of the data set for the HG-U133 Plus2 GeneChip (Affymetrix) were analyzed using GeneSpring and GSEA (Figure 4), and the results demonstrated that the expression levels of the oligomer probes of RhoA (Figure 4A), SREBF1, and SCD (Figure 4B) were upregulated in patients with ccRCC. Additionally, their related pathways including Rho protein signal transduction, focal adhesion (Figure 4A), and unsaturated fatty acid biosynthesis (Figure 4B) were upregulated in ccRCC versus normal renal tissues, as revealed by GSEA.

### 2.4. Mechanism Underlying the Induction of RhoA by Simvastatin

Although simvastatin serves as an HMG-CoA inhibitor for inhibiting Rho prenylation, RhoA is upregulated in ccRCC cells treated with simvastatin. Thus, the cytosol–membranous partition of RhoA was examined (Figure 5A); the analysis revealed that the RhoA protein was mainly present in the cytosol but not the plasma membrane, indicating that the majority of the induced RhoA was not activated in simvastatin-treated Caki2 cells. We also investigated the mechanism underlying the induction of RhoA protein by simvastatin; the results (Figure 5B) indicated that the upregulation of RhoA by simvastatin is attributable to transcriptional and translational regulation based on the observation that actinomycin and cycloheximide can mitigate simvastatin-mediated RhoA induction. SB202380, a p38MAPK inhibitor and a proteasome activator [10], was used to examine whether proteasome activity can reduce the RhoA levels upregulated by simvastatin. However, no apparent alteration was observed on the incubation of simvastatin-treated ccRCC cells with SB202380. Additionally, simvastatin treatment induced ER stress, as indicated by the increased IRE1α levels (an ER stress marker) in ccRCC cells (Figure 5C). Whether simvastatin-mediated autophagy and ER stress contribute to RhoA accumulation was investigated through treatment with their respective inhibitors (e.g., 3-MA and 4-PBA); the results in Figure 5C reveal that the inhibition of both autophagy and ER stress can attenuate the RhoA accumulation caused by simvastatin. Additionally, the activity of RhoA was examined by the RhoA-GTP antibody in Western blot analysis. Simvastatin significantly reduced the activated form of RhoA; however, the employed inhibitors shown in Figure 5B,C were not apparently able to rescue the effect of simvastatin on RhoA inactivation.

Moreover, the effects of various RhoA activities on the morphology, migration, and malignant features of ccRCC cells were evaluated in cells treated with Y27632 (a ROCK inhibitor) or cells overexpressing DN- and CARhoA. The results (Figure 6A) demonstrated that cells treated with simvastatin and Y27632 or cells overexpressing DNRhoA displayed prolonged cell morphology, with an increased length/width axis compared with control and CARhoA cells. By contrast, control and CARhoA cells exhibited more stress fiber formation, as revealed by rhodamine phalloidin staining, than cells treated with simvastatin or Y27632 or cells overexpressing DNRhoA. Similarly, the antimigratory effect of simvastatin was mimicked in cells overexpressing DNRhoA or treated with Y27632, whereas cells overexpressing CARhoA and supplemented with mevalonate exhibited enhanced migration compared with control cells (Figure 6B). We previously demonstrated that simvastatin increased RhoA expression. Thus, the RhoA activity of cell lysates treated with simvastatin was examined by evaluating ROCK activity, a downstream target of RhoA, through ELISA analysis. The results shown in Figure 6C indicate that simvastatin reduced ROCK activity (0.055 mU) to a similar extent as that in cells expressing DNRhoA (0.05 mU) and in Caki2 cells (0.07 mU), whereas CARhoA cells exhibited increased ROCK activity (0.087 mU).

The effects of various RhoA activities on the malignant features of ccRCC cells were assessed through Western blot analysis. Cells overexpressing DNRhoA exhibited higher expression levels of pAMPKα and LC3B II/I and smaller reductions in the levels of pAKT1, matrix metalloproteinase-2 (MMP2), and phospho-translational repressor protein 4E-binding protein 1 (p4EBP1) compared with CARhoA cells. Additionally, RhoA activity was validated using various markers including LIM domain kinase 2 (LIMK2), phospho-myosin light chain (pMLC), and the membrane-bound form of RhoA after cytosolic–membrane fractionation (Figure 6D); the analysis revealed RhoA induction in CARhoA but not DNRhoA cells. The effects of DNRhoA and CARhoA on ccRCC viability were also investigated using an MTT assay. DNRhoA cells exhibited a longer doubling time and reduced cell viability compared with control cells, whereas CARhoA cells exhibited a higher rate of cell proliferation (Figure 6E).

## 3. Discussion

Effective long-term treatments for advanced and metastatic ccRCC are lacking because the reduction of cell multiplication is not efficient in attenuating tumor progression and metastasis. Therefore, in addition to the tumoricidal effect, this study also investigated the anti-ccRCC effects of simvastatin on lipid metabolism and cell invasion.

Statins (e.g., simvastatin and lovastatin) are clinically used as lipid-lowering medications. Accumulating evidence indicates that statins can inhibit tumor growth and induce apoptosis in various tumor cell lines [11,12]. Statin use in targeted therapy is associated with an increased survival rate among patients with metastatic RCC [13] and a reduced risk of progression and overall mortality after surgery for localized RCC [14]. However, cohort studies have reported conflicting results regarding the therapeutic effectiveness of statins in RCC [15]. A recent systematic review and meta-analysis demonstrated that statins significantly improved cancer-specific and overall survival among patients with kidney cancer [16]. The reduced RhoA activity and lipid accumulation might further increase the antitumor effects of simvastatin by suppressing cell metastasis and therapeutic resistance. The results of this study demonstrated that simvastatin caused cell cycle arrest in the G0/G1 phase, at least in part, by increasing p21/27 and reducing cyclin D1/D3 expression (Figure 2E). In addition to decreasing cell proliferation, simvastatin induced autophagy and cell apoptosis (Figure 2 and Figure 3). Autophagy may be initiated by cell stress, which would eventually lead to cell death. Notably, we observed that simvastatin significantly induced LKB1 and AMPKα activation through phosphorylation and reduced AKT phosphorylation. The activation of pLKB1/pAMPKα contributes to pmTOR inhibition, autophagy induction, and reduced cell migration. Notably, the alterations of key proteins involved in the aforementioned events, including p21/p27, cyclin D1/D3, pLKB1/pAMPKα, and SCD/FASN, could be reversed by mevalonate supplementation, indicating that simvastatin acts by inhibiting the mevalonate pathway. Thus, simvastatin treatment exerts its antitumor effects through cell cycle arrest and its antimigration and tumoricidal effects on ccRCC cells through the inhibition of the mevalonate pathway.

To recapitulate our findings in patients with ccRCC, we performed a GSEA analysis of the human ccRCC GSE53757 data set for the lipid metabolism and cell migration/RhoA pathway and compared the related gene expression profiles with that of the paired normal renal tissues in a GeneSpring analysis. We found that the expression levels of SREBF1 and SCD were significantly increased in patients with ccRCC across the four TNM stages. Lipid accumulation has been reported to protect cancer cells from chemotherapy through ROS scavenging; therefore, a reduction in lipid accumulation can increase treatment efficacy. We demonstrated that simvastatin could reduce lipid deposition in ccRCC cells by correcting metabolic abnormalities. We hypothesize that the activation of the LKB1/AMPKα pathway is essential to reducing lipid deposition in simvastatin-treated ccRCC cells. However, the underlying mechanism is currently under investigation in the laboratory.

We also observed that simvastatin reduced cell migration, which was associated with decreased RhoA activity, but it induced RhoA expression in ccRCC cells. This phenomenon is attributable to a negative feedback mechanism caused by the inhibition of RhoA prenylation by simvastatin. Although RhoA was transcriptionally and translationally upregulated by simvastatin, RhoA was confined to the cytosol because protein prenylation was reduced in cytosolic–membrane fractionation (Figure 5). Additionally, simvastatin increased ER stress, accompanied by increased RhoA accumulation, which can be eliminated by 4-PBA (an ER stress inhibitor) in simvastatin-treated ccRCC cells. Moreover, simvastatin reduced RhoA activity, which was further validated by an ELISA analysis of the activity of ROCK (a downstream target of RhoA) (Figure 6C). The results revealed that simvastatin reduced RhoA activity. We also found that the expression of the RhoA protein and its related pathway were upregulated in the ccRCC data set. These results validated the findings of the GeneSpring analysis of clinical data, which indicated markedly increased RhoA levels in ccRCC of TNM stages 1–2 compared with the levels in ccRCC of stages 3 and 4 versus their paired normal tissues. Thus, early treatment with simvastatin might inhibit the progression of ccRCC by blocking RhoA activation. Thus, RhoA inactivation by simvastatin treatment might have led to the reduced migration of ccRCC cells in the transwell assay, which was confirmed in DNRhoA, but not CARhoA, cells.

Statins have been reported to induce autophagy in cancer cells, which in part contributes to the antimetastatic effect of statins, presumably by interfering with the posttranslational modification of Rho GTPases [17]. Belaid et al. [18] demonstrated that cell autophagy plays a critical role in the degradation of active RhoA. Autophagy serves as a RhoA regulator, which aids in maintaining an appropriate amount of active RhoA in cells in order to prevent tumor metastasis. In the present study, we observed the apparent induction of RhoA, pLKB1, and pAMPKα in simvastatin-treated ccRCC cells. Although the antimigratory effects of simvastatin were induced by RhoA inactivation by simvastatin, the mechanism by which the LKB1/AMPKα pathway causes reduced cell migration warrants further investigation. The induction of pLKB1/pAMPKα has been reported to contribute to pmTOR inhibition, autophagy, and reduced cell migration [19]. Our findings are supported by a study that reported that LKB1 deficiency disturbs the polarity of mammary epithelial cells, resulting in cell disorder and the increased invasion and migration of epithelial cells [20]. The activation of key targets, pLKB1 and pAMPKα, by simvastatin was significantly reversed by mevalonate supplementation (Figure 3A). RhoA upregulation was also dependent on the effect of simvastatin on HMG-CoA reductase inhibition, as evidenced by its significant reversal in cells with mevalonate supplementation.

## 4. Materials and Methods

### 4.1. Agents

Chemicals were purchased from the following sources: simvastatin, cycloheximide, actinomycin D, and 4-PBA from Sigma-Aldrich (St. Louis, MO, USA); 3-methyladenine (3-MA) from Merck Millipore (Darmstadt, Germany); and 3-(4,5-dimethylthiazol-2-yl)-2,5-diphenyltetrazolium bromide (MTT) from SERVA Electrophoresis GmbH (Berlin, Germany). SB202380 was purchased from Tocris Cookson Inc. (Bristol, UK), and mevalonate was purchased from Biosynth Carbosynth (Berkshire, UK). Unless specified, stock solutions were prepared by dissolving powdered drugs in dimethyl sulfoxide. Protein assay agents were purchased from Bio-Rad (Hercules, CA, USA). The chemical concentrations of agents and treatment duration for each assay were set according to our previous publications [21] or pilot studies.

### 4.2. Cell Culture Systems and Subcutaneous Transplantation of Caki2 Cells in Nude Mice

ccRCC cells of Caki1, Caki2, and 786O were maintained in RPMI medium and A498 cells in minimum essential medium (MEM) supplemented with 10% fetal bovine serum (FBS). ccRCC cells were all purchased from ATCC, which were grown to 85–95% confluence before use. RPMI-1640 medium, MEM, FBS, and tissue culture reagents were obtained from Invitrogen (Carlsbad, CA, USA). Cells from passages 5 to 20 were used in the experiments. All animal study procedures were conducted in accordance with the Taipei Medical University Animal Care and Use rules (licenses No. LAC-2018-0436) and Institutional Animal Care and Use Committee or Panel (IACUC/IACUP). Eight-week-old male Balb/c nude mice weighing 20–25 g were maintained in a laminar air-flow cabinet in specific pathogen-free facilities in the Animal Center of Taipei Medical University. After anesthetization, the skins close to thigh and abdomen at the back of the mice were lifted to separate them from the underlying muscle, and then mice were subcutaneously injected with the 5 × 10^8^ viable tumor cells suspended in 0.1 mL of 1× HBSS via a 29-gauge needle. The growth of implanted ccRCC cells at subcutaneous tissue were measured weekly, and tumor volume was calculated as follows [22]: tumor volume = (length × width^2^)/2. Tumor weight values were plotted day by day post-injection, and tumor doubling time was calculated from the resulting graph. When the tumor size reached 2 cm in diameter, the mice were sacrificed. Stock solution of 1 mg/kg simvastatin was prepared by dissolving in DMSO. 

### 4.3. Analysis of Cell Cycle Regulation and Apoptosis through Flow Cytometry

For cell cycle analysis, cells were seeded in 60 mm culture dishes and then serum-deprived for 24 h, followed by the indicated treatment. Harvested cells were fixed with 70% ethanol and allowed to set for at least 1 h. To remove ethanol, cells were centrifuged and washed with phosphate-buffered saline (PBS) twice. DNA underwent propidium iodide (PI) staining (PI: 20 µg/mL, Triton-X 100: 0.1%, RNase A: 0.2 µg/mL in ddH2O) for at least 30 min in the dark. Additionally, apoptosis induced by simvastatin in ccRCC cells was detected through annexin V-FITC/PI double labeling flow analysis according to the manufacturer’s instructions (Elabscience, Houston, TX, USA). Flow cytometry was performed using CytoFLEX and CytoExpert software (Beckman Coulter); dot plots and histograms were used for data visualization. Additionally, following two washes with PBS, simvastatin-treated ccRCC cells were stained with 1 μg/mL 4′,6-diamidino-2-phenylindole (DAPI) dye in PBS for 8 min for evaluating apoptotic bodies. Fluorescence was analyzed using a Zeiss Axio Observer Z1 inverted phase contrast microscope (Wilmington, MA, USA).

### 4.4. Cell Migration

Transwell in vitro invasion assays were conducted to evaluate the transmigration of cells, as described previously [23], by using 24-well transwell units with 8 μm polyvinylpyrrolidone-free polycarbonate filters (Dow Corning, Midland, MI, USA) that had been coated with 1 mg/mL collagen for 30 min at 37 °C. Pretreated cells (2.5 × 10^4^ cells) were seeded in medium containing 2% FBS in the upper compartment of the transwell plates. Medium containing 10% FBS was added to the lower compartment. The transwell plates were incubated at 37 °C for 12 h in a humidified atmosphere containing 5% CO_2_. Cells that migrated to the lower surface of the membrane were observed through staining with crystal violet dye under a light microscope. Cells migrating from the leading edge were photographed and counted at 0 and 16 h by using an Olympus SC-5 CCD camera (Tokyo, Japan) attached to an Olympus BX51 microscope system. Fluorescence was observed using a charge-coupled device camera (DP72, Olympus, Melville, NY, USA) attached to a microscope system (BX51, Olympus) at 200× magnification. Four slides with coverslips were examined in each experimental group. The subsequent procedures were described in a previous study [24].

### 4.5. Oil-Red O Staining

Cells were seeded in 24-well cell culture plates and treated with the drugs of interest. Cells were fixed in 4% paraformaldehyde for at least 30 min. During fixation, oil-red O (ORO) stock solution (3 mg ORO powder in 1 mL isopropanol) was diluted with deionized water (ddH2O) at a dilution rate of 3:2, and the staining solution was filtered. The fixing solution was discarded, and cells were washed with ddH2O once. Isopropanol (60%) was added to evaporate the remaining water. Lipids were stained using ORO staining solution for 10 min. Wells were washed with tap water until the water ran clear. ORO was dissolved by adding 99% isopropanol, and absorbance was measured using a spectrophotometer at 500 nm.

### 4.6. Preparation of Cell Fractions (Cytosol and Membrane) and Western Blot Analysis

Cells grown in 6 or 10 cm^2^ dishes were harvested after the indicated treatments with protease inhibitors. To prepare membrane–cytosolic fractions, cells were collected after the indicated treatments and incubated in 0.1 mL of hypotonic buffer (10 mM Tris (pH 7.5), 0.5 mM EDTA, and 2 mM phenylmethylsulfonyl fluoride) at 4 °C for 30 min. After centrifugation, the supernatant (cytosolic fraction) was collected, and the pellet was resuspended in 0.1 mL of radioimmunoprecipitation assay buffer and incubated at 4 °C for 30 min. The resulting fractions were sheared 100 times through an insulin syringe with a 29-G needle. After centrifugation, the supernatant (membrane fraction) was collected for analysis. Western blotting was performed as described previously [24], and the following antibodies were used: antibodies against the RhoA, pAKT1/AKT1, pmTOR/mTOR, p4EBP1, pMLC, p27, FASN, SCD PARP, α-tubulin, β-actin, and pan-cadherin (Santa Cruz Biotechnology, Dallas, TX, USA); LC3BI/II, CDK4, and cyclin D3 (GeneTex, Irvine, CA, USA); p21 (BD Biosciences, San Jose, CA, USA); and cyclin D1, pLKB1, and pAMPKα/AMPKα, (Cell Signaling, Danvers, MA, USA); RhoA-GTP (NewEast Biosciences, King of Prussia, PA, USA). Total protein (80 μg) was separated on a 10% acrylamide gel through sodium dodecyl sulfate–polyacrylamide gel electrophoresis. Hybond-P membranes (GE Healthcare Life Sciences, Waukesha, WI, USA) containing the electrophoretically transferred protein bands were probed using various primary antibodies. Band intensities were normalized to the band intensity of α-tubulin/β-actin (control) by using an IS-1000 digital imaging system (ARRB, Victoria, Australia).

### 4.7. Cells Overexpressing RhoA Variants and Activity Assay

RhoA constructs of constitutively active (CA) RhoA (G14V) and dominantly negative (DN) RhoA (T19N) in pUSEamp were purchased from Millipore (Burlington, MA, USA). Caki2 cells were transfected with pUSEamp-overexpressing variants (6 µg/10 cm Petri dish) overnight by using Lipofectamine 2000 (Invitrogen). After transfection, cells were plated in RPMI supplemented with 10% FBS and 400 µg/mL G418 as selective pressure. G418-resistant cells were selected and expanded. The resulting cells were harvested and subjected to a Western blot analysis of the proteins of interest. ROCK activity was assessed using enzyme-linked immunosorbent assay (ELISA) kits (Millipore, Burlington, MA, USA) according to the manufacturer’s protocol.

### 4.8. Gene Set Enrichment Analysis

Expression profiles of the gene expression data series (GSE) 53757 from the Gene Expression Omnibus (GEO) database at the National Center for Biotechnology Information were selected and analyzed using the HG-U133_Plus_2 GeneChip (Affymetrix) [25]. GeneSpring (v.14.9) and GSEA (v4.2) were employed to normalize the data and to identify gene sets enriched in the ccRCC data set compared with paired adjacent normal renal tissues, respectively. This data set contains 142 expression data files from normal and renal tumor tissues of 71 patients with ccRCC, grouped into tumor–node–metastasis (TNM) stages 1–4. GSEA was performed using GSEA v4.2 software. After gene set permutations were practiced 1000 times, significant gene sets were identified based on enrichment scores (ES) and false discovery rates (FDRs) below 0.25. All gene set files used were obtained from the Molecular Signatures Database.

### 4.9. Statistical Analysis

Data are presented as the mean ± SEM of at least three experiments. Differences in the means of two groups were calculated using unpaired Student’s *t* tests, and differences among multiple groups were determined using one-way analysis of variance. The Bonferroni method was used for post hoc analysis. *p* < 0.05 was considered significant.

## 5. Conclusions

We revealed that simvastatin reduces cell viability and increases autophagy induction and apoptosis. In addition, it reduces cell metastasis and lipid accumulation, the target proteins of which can be reversed through mevalonate supplementation. Simvastatin significantly reduces RhoA activity, accompanied by the reduced cell migration, cell proliferation, and malignancy of ccRCC cells, thus suggesting the effectiveness of simvastatin in ccRCC treatment. Considering its potential antimigratory effect, we anticipate that simvastatin may be an effective therapeutic strategy for advanced ccRCC. The use of statins in chemotherapy regimens warrants further evaluation because statins are commonly used in the treatment of cardiovascular disease in some elderly patients. The mechanism underlying the beneficial antitumor effects of simvastatin revealed in this study might provide strong evidence for further clinical investigations.

## Figures and Tables

**Figure 1 ijms-24-09738-f001:**
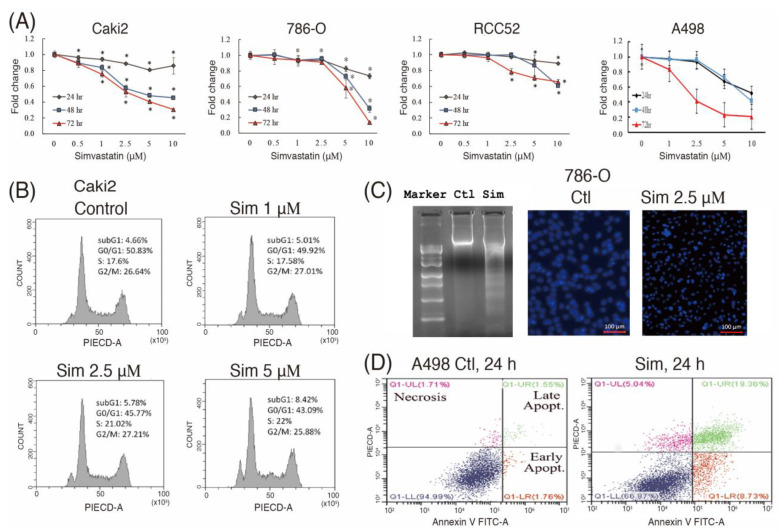
Simvastatin inhibited cell viability in various ccRCC cells in a concentration-dependent manner. (**A**) The effect of simvastatin treatment on cell viability was analyzed using the MTT assay. A total of 1.5 × 10^4^ cells/well were seeded in a 24-well plate, which was followed by 24–72 h of simvastatin challenge. (**B**) Cells were incubated in 0.5% serum RPMI for 24 h to induce quiescence, which was followed by indicated concentrations of simvastatin treatment for 24 h in 10% FBS RPMI medium. The percentages of cells in the subG1 phase and the cell cycle were analyzed through flow cytometry analysis. The number of cells is plotted on the *y*-axis and PI energy-coupled dye labeling (DNA content) on the *x*-axis. Results of a representative experiment are shown. (**C**) The effect of simvastatin on cell DNA fragmentation and apoptosis was evaluated by electrophoresis and DAPI staining. Images were taken under 200× magnification. (**D**) Cells pretreated with simvastatin overnight were treated with PFOS for 24 h. Harvested cells were double-stained with annexin V-PI, and apoptosis was assessed through flow cytometry. Results are representative of three independent experiments. Data are representative of the results of three independent experiments, and the data are presented as the mean ± SEM (* *p* < 0.05 versus control).

**Figure 2 ijms-24-09738-f002:**
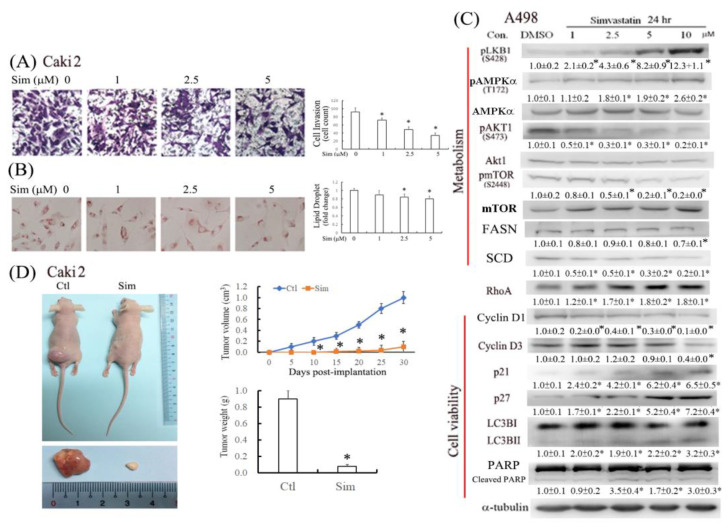
Reduction of cell migration and lipid deposition in simvastatin-treated ccRCC cells. (**A**) Cells were seeded in a nutrient gradient medium containing 2.5 μM simvastatin and incubated for 16 h, followed by crystal violet staining. Microscopic observation and the quantification result of migrated Caki2 cells. Images were taken under 200× magnification. (**B**) Cells were treated with simvastatin and then fixed with paraformaldehyde and stained with oil-red O (ORO) stain for 10 min. The stained cells were then dissolved in isopropanol, and absorbance was detected using a spectrophotometer. Microscopic observation of Caki2 cells treated with simvastatin at different concentrations for 24 h (200× magnification) after ORO staining and quantification of (**B**). * *p* < 0.05 versus control group. (**C**) Protein expression pattern of the concentration-dependent effects of simvastatin on malignant features of A498 cells. Cells were treated with increasing concentrations (1–10 μM) of simvastatin, and the expression levels of the indicated proteins involved in the cell cycle, cell viability, and malignant features of ccRCC were analyzed through Western blotting. The intensity of each protein band was quantitated through densitometry and normalized with an internal control of α-tubulin. Results are presented as the mean ± SEM for each membrane blot (* *p* < 0.05 versus control). Data are representative of the results of three independent experiments. (**D**) After subcutaneous xenografts of ccRCC cells in nude mice in 2 days, simvastatin (1 mg/kg body weight) was intraperitoneally given to each group twice a week for 30 days, in which tumor sizes and volumes were evaluated. Data are presented as the mean ± SEM. * *p* < 0.05, compared with vector alone (*n* = 3 for each group).

**Figure 3 ijms-24-09738-f003:**
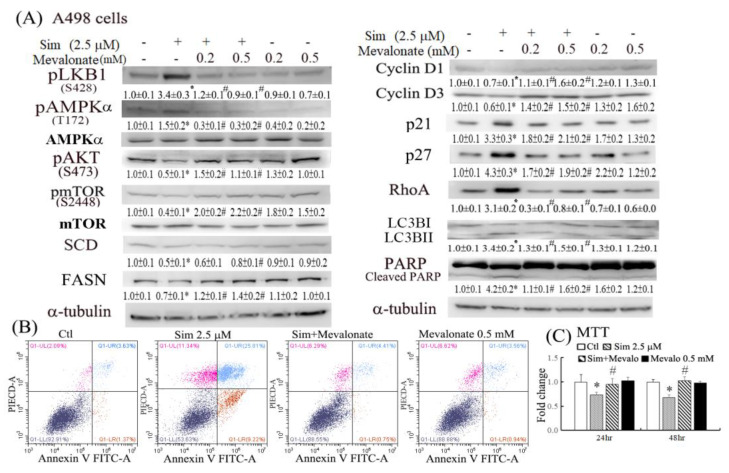
Reversal of the anti-ccRCC effects of simvastatin by mevalonate supplementation in terms of the signaling of the bioenergetic metabolism and apoptosis in ccRCC cells. (**A**) A498 cells were pretreated with mevalonate (0.2 and 0.5 mM) for 10 min, followed by a 24 h treatment with simvastatin. Western blotting analysis was performed to analyze the proteins involved in bioenergetic metabolism, cytoskeletal organization (e.g., RhoA), cell cycle regulation (e.g., cyclin D1/D3), and cell apoptosis (e.g., cleaved PARP). The intensity of each protein band was quantitated through densitometry and normalized with an internal control of α-tubulin. Results are presented as the mean ± SEM for each membrane blot (* *p* < 0.05 versus control; # *p* < 0.05 versus simvastatin). Data are representative of the results of three independent experiments. (**B**) Cells treated with 2.5 μM simvastatin for 24 h were stained with annexin V-FITC/PI and assayed for apoptosis through flow cytometry. (**C**) A498 cells were pretreated with 0.5 mM mevalonate for 10 min, followed by a 6−24 h treatment with 2.5 μM simvastatin, and cell viability assessments were conducted using MTT assays. * *p* < 0.05 versus control groups; # *p* < 0.05 versus simvastatin.

**Figure 4 ijms-24-09738-f004:**
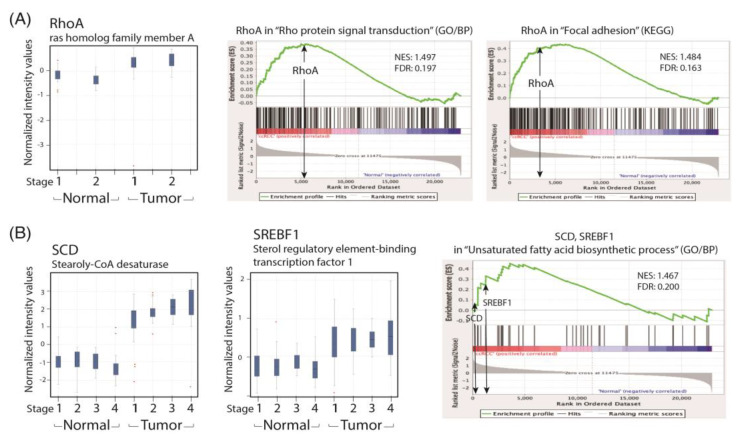
GeneSpring and GSEA analysis of cell migration and metabolism pathways in the human ccRCC data set. (**A**) A box and whisker plot of RhoA gene in renal tumor versus normal tissues is displayed for patients in TNM stages 1–2. GSEA plots of Rho protein signal transduction and focal adhesion pathways are shown; FDR < 0.25 indicates a significant difference in a pathway when comparing ccRCC tumor tissues to control. (**B**) Box and whisker plots of SREBF1 and SCD genes and a GSEA plot of unsaturated fatty acid biosynthesis processes between normal versus renal tumor tissues are displayed for patients in TNM stages 1−4.

**Figure 5 ijms-24-09738-f005:**
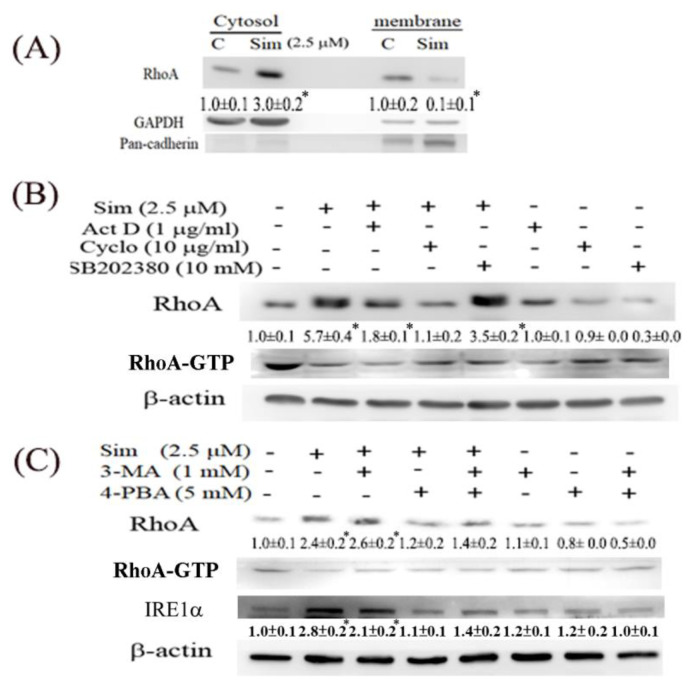
Mechanism underlying the fractionation and induction of RhoA by simvastatin in A498 cells. (**A**) Effect of simvastatin on cytosolic-membranous translocation of RhoA protein after fractionation, with GAPDH and pan-cadherin as respective input controls. (**B**) Cells were pretreated with actinomycin D (ActD), cycloheximide (CHX), and SB202380, inhibitors of transcription, translation, and proteasome, respectively, for 1 h, followed by simvastatin treatment for 24 h. (**C**) Cells were pretreated with 3-MA and 4-PBA, inhibitors of autophagy and ER stress, for 1 h, followed by simvastatin treatment for 24 h. The intensity of each protein band was quantitated through densitometry and normalized with an internal control of GAPDH or β-actin. Results are presented as the mean ± SEM for each blot (* *p* < 0.05 versus control). Data are representative of the results of three independent experiments.

**Figure 6 ijms-24-09738-f006:**
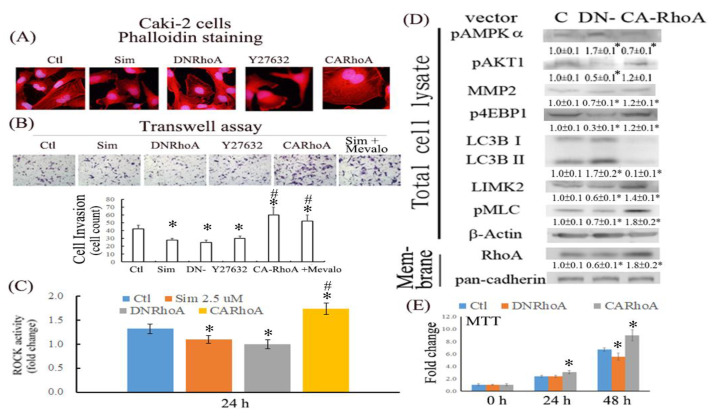
Effects of RhoA activity on alleviating ccRCC progression. (**A**) The morphology of Caki2 cells subjected to the indicated treatments was investigated after rhodamine phalloidin staining to evaluate actin stress fiber formation through fluorescent microscopy. Images were taken under 400× magnification. (**B**) Cells treated with simvastatin or those overexpressing CA- or DNRhoA were assessed for cell migration after 16 h of simvastatin treatment. Images were taken under 100× magnification (* *p* < 0.05 versus control; # *p* < 0.05 versus simvastatin). (**C**) Cell lysates of cells treated with simvastatin for 24 h or those overexpressing CA- or DNRhoA were harvested and analyzed for ROCK activity by using the ELISA assay. The bar graph depicts the fold change in ROCK activity compared with the control group (* *p* < 0.05 versus control; # *p* < 0.05 versus simvastatin). (**D**) With respect to RhoA activity, the altered levels of pAMPKα, pAKT1/2, MMP2, p4EBP1, and LC3BII/I were evaluated using a Western blot analysis, and the results were validated using markers including LIMK2, pMLC, and membrane-bound RhoA, with pan-cadherin as an input control. The intensity of each protein band was quantitated through densitometry and normalized with an internal control of β-actin. Results are presented as the mean ± SEM for each membrane blot (* *p* < 0.05 versus control). (**E**) Cell viability of various Caki2 cells was examined at 1 and 2 days after treatment by using MTT assays (* *p* < 0.05 versus empty vector at each respective day). Data are representative of the results of six independent experiments, and the data are presented as the mean ± SEM (* *p* < 0.05 versus control).

## Data Availability

Raw data of Western blot analysis are included in the Appendix A. Gene expression profiles of ccRCC are downloaded from open-access GEO/NCBI (https://www.ncbi.nlm.nih.gov/geo/ accessed on 20 December 2018) with accession code of GSE53757.

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
