# Peer review of "Tumoricidal Activity of Simvastatin in Synergy with RhoA Inactivation in Antimigration of Clear Cell Renal Cell Carcinoma Cells"

_ijms, 2023, doi:10.3390/ijms24119738_

Round 1

Reviewer 1 Report

The article Yuan-Chii Gladys Lee et al is devoted to treatment of cell renal cell carcinoma by  Simvastatin, a lipid-lowering drug with reduced mevalonate synthesis. Authors showed that Simvastatin reduce cell viability and increase autophagy induction, apoptosis, reduced cell metastasis and lipid accumulation in cell renal cell carcinoma cell lines. Moreover, authors showed molecular mechanism of Simvastatin which is depending on reduction of cancer metastasis by suppressing the RhoA pathway.

However, there are one major comment: The quality of the figure is not good.

And there are several minor comments:

Figure 1

Quality of the figure is not good. Cell cycle and annexinV are very bad resolution. It is hard to understand numbers (please choose another colour).

According to the picture the effect of DNA fragmentation is hard to make a conclusion. Please, use DNA fragmentation assay (DNA electrophoresis).

Figure2

1.     It is important to show the level of expression also total AMPK and mTOR in western blots

2.     Why the levels of cleavage PARP1 is too small after treatment of Simvastatin?

Figure 3

1.     It is important to show the level of expression also total AMPK and mTOR in western blots

2.     Please choose another colour for annexinV assay

Figure 5

1.     Please, use specific antibodies for activation RhoA (RhoA-GTF).

2.     Also to show that inhibitor of RhoA (Y27632) is working (also using antibody for RhoA-GTF)

Author Response

We thank you for your invaluable assistance in revising this manuscript. We have closely reviewed your comments and have edited the manuscript accordingly.

Reviewers' comments:

Reviewer's Responses to Questions

The article Yuan-Chii Gladys Lee et al is devoted to treatment of cell renal cell carcinoma by Simvastatin, a lipid-lowering drug with reduced mevalonate synthesis. Authors showed that Simvastatin reduce cell viability and increase autophagy induction, apoptosis, reduced cell metastasis and lipid accumulation in cell renal cell carcinoma cell lines. Moreover, authors showed molecular mechanism of Simvastatin which is depending on reduction of cancer metastasis by suppressing the RhoA pathway.

However, there are one major comment: The quality of the figure is not good.

And there are several minor comments:

Figure 1

Quality of the figure is not good. Cell cycle and annexinV are very bad resolution. It is hard to understand numbers (please choose another colour).

According to the picture the effect of DNA fragmentation is hard to make a conclusion. Please, use DNA fragmentation assay (DNA electrophoresis).

Answer: To improve the quality of Figure 1, the font size for the percentage of cell cycle has been increased and the color legend of Annexin V has been labelled by different color with better contrast as shown in the revised Fig. 1. Additionally, in accordance with the reviewer’s concerns, we have added the results of DNA fragmentation assay (DNA electrophoresis) in the revised Fig. 1C to validate the apoptotic cell death in cells treated with simvastatin.

Figure2

  1. It is important to show the level of expression also total AMPK and mTOR in western blots

Answer: The expression levels of total AMPK and mTOR in western blot analysis have been included in the revised version of the figure 2.

  1. Why the levels of cleavage PARP1 is too small after treatment of Simvastatin?

Answer: The cleaved form of PARP1 exhibited time-dependent fashion; more dramatic increases occurred after day 2 and day 3 of simvastatin treatment. Although it was obvious at 24 h, the original uncleaved form of PARP1 is quite abundant in this chosen ccRCC cell types.

Figure 3

It is important to show the level of expression also total AMPK and mTOR in western blots. Please choose another colour for annexin V assay.

Answer: The expression levels of total AMPK and mTOR in western blot analysis have been included in the revised version of the figure 3A. Additionally, the percent of annexin V staining in the flow cytometry has been labelled by different color with better contrast in the revised version of Figure 3B.

Figure 5

  1. Please, use specific antibodies for activation RhoA (RhoA-GTP).
  2. Also to show that inhibitor of RhoA (Y27632) is working (also using antibody for RhoA-GTF)

Answer: In accordance with the reviewer’s concerns, we have added the results of western blot analysis of RhoA-GTP in the revised Fig. 5 to validate the activated RhoA; the activity of RhoA was examined by RhoA-GTP antibody in Western blot analysis. Simvastatin significantly reduced activated from of RhoA, however, the employed inhibitors in Figs. 5B and 5C were not apparently to rescue the effect of simvastatin on RhoA inactivation. (see p6, lines 215-8 of the revised manuscript). Furthermore, we need to correct our mistake about Y27632 that is a Rock inhibitor other than a RhoA inhibitor as corrected in the revised manuscript (see p7, lines 231-2).

Reviewer 2 Report

The authors treated four renal carcinoma cell lines with different concentration of simvastatin that decreased cell proliferation, increased autophagy and induced apoptosis. Simvastatin also reduced metastasis and lipid accumulation. Cholesterol synthesis and protein prenylation are drastically reduced in treated cells impairing RhoA activation. Based on these finding the authors suggested that simvastatin can be used as an adjunct therapy for renal carcinoma.

Major concern.

Although the authors generated enough in vitro evidence about the effect of simvastatin on renal carcinoma cells, it will be crucial to show its effect on tumor growth and potential pathological effects on mice’s organs (heart, kidney and liver).  Thus, the authors must induce tumors in immunocompromised mice using one of the renal carcinoma cell lines and measure tumor volume in mice treated with simvastatin and mice treated with vehicle.  

Minor concern.

Figure 1B and 4A  have poor resolution and legends are difficult to read .

Author Response

We thank you for your invaluable assistance in revising this manuscript. We have closely reviewed your comments and have edited the manuscript accordingly.

The authors treated four renal carcinoma cell lines with different concentration of simvastatin that decreased cell proliferation, increased autophagy and induced apoptosis. Simvastatin also reduced metastasis and lipid accumulation. Cholesterol synthesis and protein prenylation are drastically reduced in treated cells impairing RhoA activation. Based on these finding the authors suggested that simvastatin can be used as an adjunct therapy for renal carcinoma.

Major concern.

Although the authors generated enough in vitro evidence about the effect of simvastatin on renal carcinoma cells, it will be crucial to show its effect on tumor growth and potential pathological effects on mice’s organs (heart, kidney and liver).  Thus, the authors must induce tumors in immunocompromised mice using one of the renal carcinoma cell lines and measure tumor volume in mice treated with simvastatin and mice treated with vehicle.  

Answer: In response to the reviewer’s suggestion, we employed xenotransplantation of Caki2 cells into Balb/c nude mice to examine the anti-ccRCC effect of simvastatin and monitor tumor growth. To assess the efficacy of simvastatin in vivo, simvastatin was given twice a week for 4 weeks to nude mice with xenograft transplantation. The results in Fig. 2D show that tumor weights (gw) and volumes (cm3) of subcutaneous xenografts grown from Caki2 cells were significantly smaller than those from additional simvastatin treatment (P < 0.05). The values of BUN in control and simvastatin groups were 18±2 and 20±2 mg/dl, and those of alanine transaminase (AST/GPT) were 45±3 and 47±2 U/L, respectively, which were within the normal ranges and no statistical significance between groups (see p4, lines 123–9, of the revised manuscript).

Minor concern.

Figure 1B and 4A have poor resolution and legends are difficult to read.

Answer: The resolution of figures and the color of legends has been amended for better presentation and easy reading in the revised Figs 1B and 4A.

Round 2

Reviewer 1 Report

The article can be accepted

Reviewer 2 Report

The authors nicely presented in vivo tumor growth data showing that simvastatin delays renal tumor growth induced in immunodeficient mice.